# Salicylate Sodium Suppresses Monocyte Chemoattractant Protein-1 Production by Directly Inhibiting Phosphodiesterase 3B in TNF-α-Stimulated Adipocytes

**DOI:** 10.3390/ijms24010320

**Published:** 2022-12-24

**Authors:** Xiaoyu Zhang, Yuan Gao, Zhuangzhuang Liu, Wenjing Li, Yuan Kang, Ximeng Li, Zhenlu Xu, Cheng Peng, Yun Qi

**Affiliations:** 1Institute of Medicinal Plant Development, Chinese Academy of Medical Sciences & Peking Union Medical College, Beijing 100193, China; 2School of Pharmacy, Chengdu University of Traditional Chinese Medicine, Chengdu 610075, China

**Keywords:** cAMP, MKP-1, p-ERK, p-p38, metaflammation

## Abstract

As a worldwide health issue, obesity is associated with the infiltration of monocytes/macrophages into the adipose tissue causing unresolved inflammation. Monocyte chemoattractant protein-1 (MCP-1) exerts a crucial effect on obesity-related monocytes/macrophages infiltration. Clinically, aspirin and salsalate are beneficial for the treatment of metabolic diseases in which adipose tissue inflammation plays an essential role. Herein, we investigated the effect and precise mechanism of their active metabolite salicylate on TNF-α-elevated MCP-1 in adipocytes. The results indicated that salicylate sodium (SAS) could lower the level of MCP-1 in TNF-α-stimulated adipocytes, which resulted from a previously unrecognized target phosphodiesterase (PDE), 3B (PDE3B), rather than its known targets IKKβ and AMPK. The SAS directly bound to the PDE3B to inactivate it, thus elevating the intracellular cAMP level and activating PKA. Subsequently, the expression of MKP-1 was increased, which led to the decrease in p-EKR and p-p38. Both PDE3B silencing and the pharmacological inhibition of cAMP/PKA compromised the suppressive effect of SAS on MCP-1. In addition to PDE3B, the PDE3A and PDE4B activity was also inhibited by SAS. Our findings identify a previously unrecognized pathway through which SAS is capable of attenuating the inflammation of adipocytes.

## 1. Introduction

Obesity is associated with the infiltration of monocytes/macrophages into adipose tissue, which causes an unresolved inflammatory state. Persistent and low-grade inflammation in adipose tissue is a key contributor to obesity-associated metabolic complications, such as insulin resistance (IR) and type 2 diabetes [1,2].

Monocyte chemoattractant protein-1 (MCP-1), a member of the CC chemokine family, plays a crucial role in the infiltration of obesity-related monocytes/macrophages into the adipose tissue [3]. It mediates the release of monocytes from the bone marrow and generates a chemokine gradient to guide monocytes into the adipose tissue (especially visceral fat) [4]. The overexpression of MCP-1 in adipocytes leads to a significant accumulation of macrophages in the adipose tissue, while MCP-1 homozygous knockout mice exhibit the opposite phenotypes [5,6]. To our knowledge, MCP-1 can be produced by various cells, either constitutively or inductively [7]. Adipocytes can secrete MCP-1 after undergoing IR or proinflammatory cytokines (e.g., TNF-α) stimulation. IR causes adipose tissue inflammation and activates proinflammatory M1 macrophages [8], while this inflammation may be of an “unresolved” or “persistent” feature. Cross-talk between adipocytes and other cells residing in the adipose tissue may be critical for determining the tones of inflammation. For instance, a well-known inflammatory cycle in the adipose tissue is that adipocyte-derived MCP-1 enhances the infiltration of monocytes/macrophages, which can be activated by adipocyte-derived free fatty acid to secrete TNF-α. In turn, the released TNF-α stimulates the adipocytes to produce more free fatty acid and MCP-1. Therefore, decreasing the MCP-1 in adipocytes may be a feasible strategy for breaking such a vicious cycle and preventing the development of obesity-associated metabolic complications.

The phosphodiesterase (PDE) superfamily contains 11 structurally related, but functionally distinct, PDE gene families (PDEs 1–11). Similarly to the cAMP- and cGMP-specific PDE, the PDE3 family consists of two members (PDE3A and PDE3B), which are generated from two similarly organized genes. In contrast to PDE3A, PDE3B is more highly expressed in insulin-sensitive cells, such as adipocytes, hepatocytes, and pancreatic β cells [9,10]. Conventionally, PDE3B is considered as a key enzyme in insulin-mediated lipogenesis, glucose uptake, and glucose transporter–4 translocation in adipocytes [11,12]. Emerging evidence has shown that PDE3B is also involved in inflammation-related events, including regulating the NLRP3 inflammasome of adipose tissue [13], mediating rheumatoid arthritis [14] and allergic airway inflammation [15].

Salicylic acid (Figure 1), a main metabolite of aspirin (Figure 1), is a plant product whose anti-inflammatory activity was first described by the famous Greek physician, Hippocrates. Unlike aspirin, salicylate does not possess the acetylating activity to inactivate cyclooxygenase in vitro [16], but both salicylate and aspirin inhibit IKKβ activation [17,18] and activate AMP-activated protein kinase (AMPK) [19]. Clinically, salicylate has been replaced by synthetic derivatives, such as aspirin and salsalate (salicylate precursors; Figure 1), which are effective for the patients with fat-induced IR or type 2 diabetes [20,21,22,23]. These clinical reports suggest that salicylate may relieve IR-related adipose tissue inflammation. Conventionally, the efficacy of salicylate on IR is thought to reflect its anti-inflammatory action via IKKβ inhibition [24]. An important question that arises from these reports is whether salicylate can break the aforementioned vicious cycle in adipose tissue inflammation through the IKKβ mechanism. In this study, we revealed a previously unrecognized target PDE3B by which salicylate sodium (SAS) suppressed MCP-1 production in TNF-α-stimulated adipocytes.

## 2. Results

### 2.1. SAS Decreases MCP-1 Production in TNF-α-Stimulated Adipocytes In Vitro

In obesity, the chemokine MCP-1 can recruit circulating monocytes into the adipose tissue where TNF-α is overproduced [25,26], which causes an unresolved inflammation. Given the clinical effects of aspirin and salsalate on metaflammation, we wondered whether their active metabolite salicylate could break the aforementioned vicious cycle in adipose tissue inflammation. To this end, we investigated the effect of SAS on MCP-1 production in TNF-α-stimulated adipocytes. The data showed that TNF-α (40 ng/mL) significantly elevated subnatant or supernatant MCP-1 in mouse primary adipocytes, as well as 3T3-L1 pre- and mature-adipocytes, while SAS significantly decreased MCP-1 production at the nontoxic concentrations (Figure 2A–C, Appendix A). Compared with the primary and mature adipocytes, the 3T3-L1 pre-adipocytes exhibited higher reactivity in response to TNF-α (≈8.4-fold increase) in the parallel experiments. Moreover, they possess proliferation capacity. Thus, 3T3-L1 pre-adipocytes were used for most of the subsequent experiments. The MCP-1 mRNA level was measured using an RT-qPCR assay. As shown in Figure 2D, TNF-α was able to significantly elevate the mRNA level of MCP-1, while the SAS concentration-dependently decreased the MCP-1 mRNA level. We also determined the effect of SAS on the mRNA stability of MCP-1. As a result, SAS did not affect the mRNA stability of MCP-1 (Figure 2E). These results suggest that the inhibitory effect of SAS on MCP-1 is attributed to suppressing its transcription.

### 2.2. SAS Decreases MCP-1 Production Independent of IKKβ Inhibition and AMPK Activation in Adipocytes

As a previous study reported that TNF-α-induced MCP-1 could be mediated by NF-κB [27], together with the fact that salicylate is an IKKβ inhibitor [17], we next evaluated whether SAS reduced MCP-1 via IKKβ inhibition. By using IKKβ siRNA, we obtained IKKβ-deficient 3T3-L1 pre-adipocytes (Figure 3A and Appendix A). Unexpectedly, the knockdown of IKKβ did not impact the effect of SAS on MCP-1 (Figure 3B). To consolidate our finding, an IKKβ inhibitor (LY2409881) was used. Unsurprisingly, it also did not decrease MCP-1 production (Appendix A).

In addition to being an IKKβ inhibitor, salicylate was also identified as an activator of AMPK [19], an attractive target for the treatment of metaflammation [28], whose activity is reduced in TNF-α-stimulated adipocytes [29]. In contrast to the previous report [30], our results indicated that AICAR, an AMPK activator, did not decrease the MCP-1 production in TNF-α-stimulated 3T3-L1 pre-adipocytes (Appendix A). Consistently, the AMPK inhibitor (Compound **C**) could not antagonize the effect of SAS (Figure 3C). These results indicate that the suppressive effect of SAS on adipocyte MCP-1 production is independent of IKKβ inhibition and AMPK activation.

### 2.3. SAS Reduces the Levels of p-p38 and p-ERK and Increases MKP-1 Expression in TNF-α-Stimulated Adipocytes

Activator protein-1 (AP-1), which is regulated by mitogen-activated protein kinases (MAPKs), including ERK, JNK, and p38, is another transcription factor that can bind to the promoter region of the MCP-1 gene to regulate MCP-1 expression. Therefore, we next assessed the effects of SAS on the phosphorylation of MAPKs (p-ERK, p-p38 and p-JNK). The results showed that the stimulation of the adipocytes with TNF-α induced the significant increase in p-ERK, p-p38 and p-JNK, while SAS could concentration-dependently decrease the levels of p-ERK and p-p38, but not p-JNK (Figure 4A–C).

MKP-1 is a dual-specificity phosphatase acting as a negative regulator of MAPKs. The down-regulation of MKP-1 is critical for the increased production of MCP-1 during the course of adipocyte hypertrophy [31]. As we know, MKP-1 is a labile protein whose half-life varies between 40 min and 2 h [32]. Indeed, our results showed that MKP-1 expression could be significantly elevated by TNF-α within 1 h and then returned to basal level at 2 h in 3T3-L1 pre-adipocytes (Figure 4D), while SAS maintained the expression of MKP-1 at 2 h after TNF-α stimulation in a concentration-dependent manner (Figure 4E).

### 2.4. SAS Decreases MCP-1 Production Depending on cAMP/PKA Pathway in Adipocytes

Given that the activation of cAMP/PKA signaling can promote MKP-1 expression [33], we next investigated the effects of SAS on the intracellular cAMP level and PKA activity. The results showed that TNF-α caused a significant reduction in the intracellular cAMP within 4 h in 3T3-L1 pre-adipocytes, while SAS could prevent this decrease in a concentration-dependent manner (Figure 5A,B). Moreover, both the cAMP antagonist (RP-cAMPS) and PKA antagonist (H89) abolished the effect of SAS on MCP-1 (Figure 5C,D). Consistently, SAS elevated the intracellular PKA activity in TNF-α-stimulated 3T3-L1 pre-adipocytes (Figure 5E). These findings demonstrated that SAS could activate cAMP/PKA signaling to inhibit MCP-1 production.

### 2.5. SAS Directly Inhibits PDE3B to Decrease MCP-1 Production in Adipocytes

The intracellular cAMP is mainly controlled by PDE3B in adipocytes [34]. The deletion of PDE3B led to a decrease of MCP-1 in the adipose tissue [13]. Does salicylate reduce MCP-1 through directly inhibiting PDE3B? To answer this question, we determined the interaction of SAS with PDE3B using a SPR assay. The results showed that SAS could directly bind to PDE3B protein with the Ka and Kd values of 8 M^−1^s^−1^ and 0.69 s^−1^, respectively (Figure 6A). Moreover, it significantly inhibited the PDE3B activity with the IC_50_ value of 13.81 mM (Figure 6B). Next, we knocked down the PDE3B of the 3T3-L1 pre-adipocytes (Figure 6C and Appendix A). In the PDE3B-deficient cells, the TNF-α-induced MCP-1 production was markedly decreased and the effect of SAS on MCP-1 almost disappeared compared with the cells transfected with scrambled siRNA (Figure 6D). Moreover, the other two PDE3 inhibitors (milrinone and cilostamide) also suppressed the MCP-1 production in the TNF-α-activated adipocytes at the concentration of 200 μM (Appendix A). These findings indicate that SAS suppresses MCP-1 production through targeting PDE3B, leading to its inhibition in adipocytes.

### 2.6. SAS Is a Non-Specific PDE3B Inhibitor

In addition to PDE3B, we also determined the effects of SAS on the other two PDE isoforms (PDE3A and PDE4B) to verify whether its inhibitory effect was selective. As shown in Figure 7, SAS also significantly suppressed both PDE3A and PDE4B activity with the IC_50_ values of 6.0 mM and 0.81 mM, respectively.

## 3. Discussion

Clinical trials showed that low-dose aspirin was able to prevent normal-weight patients from cardiovascular events or death, but for patients with obesity (≥70 kg), only a high-dose of aspirin was effective [35]. Consistently, our study also demonstrated that high-dose SAS (45 mg/kg) markedly prevented the recruitment of circulating monocytes into the adipose tissue of obese mice (Appendix A). In addition, in vitro, only high-concentration SAS (1.25–5 mM) could impede the expression of MCP-1, a key chemokine that recruits circulating monocytes into the adipose tissue [26] in adipocytes (Figure 2).

To our knowledge, MCP-1 is one of the NF-κB target genes, and the suppression of IKKβ/NF-κB-dependent inflammation is beneficial to improving IR and type 2 diabetes [20,36,37]. AMPK, a sensor or gauge of cellular energy, plays a critical role in modulating energy homeostasis by regulating the critical metabolic pathways. Adipose tissue inflammation can dampen AMPK activity [29]. Conversely, AMPK can antagonize fatty acid-induced inflammation [38], while a lack of AMPK will exacerbate IR [39]. Based on the above-mentioned background, together with the fact that salicylate can inhibit IKKβ and activate AMPK [17,19], it was taken for granted that either or both of them were the underlying mechanism(s) for the effect of SAS on MCP-1 production. Unexpectedly, the knockdown of IKKβ was unable to affect the effect of SAS (Figure 3A,B and Appendix A). Moreover, LY2409881, an IKKβ inhibitor, could not suppress MCP-1 production in TNF-α-stimulated 3T3-L1 pre-adipocytes (Appendix A). These findings were consistent with previous reports that NF-κB signaling did not affect MCP-1 production in other cells [40,41,42]. Similarly, the AMPK activator (AICAR) also did not decrease the TNF-α-induced MCP-1 production (Appendix A). These findings demonstrate that the suppressive effect of SAS on MCP-1 production in adipocytes is independent of IKKβ inhibition and AMPK activation.

Recently, we have demonstrated that MAPKs/AP-1, rather than NF-κB, is responsible for MCP-1 production in TNF-α-activated adipocytes [43]. As expected, SAS could decrease the levels of p-ERK and p-p38 (Figure 4A,B). Given that the p38 MAPK pathway was shown to regulate the stability of MCP-1 mRNA in LPS-activated THP-1 monocyte cells [44], we also determined the effect of SAS on the post-transcriptional level of MCP-1. However, the result showed that it had no impact on the mRNA stability of MCP-1 in TNF-α-stimulated adipocytes (Figure 2E). As we know, MKP-1, a nuclear tyrosine/threonine phosphatase, characterized by a transient expression pattern with rapid induction, can dephosphorylate the activated ERK and p38 [45,46]. Therefore, we inferred that the suppressive effect of SAS might be attributed to MKP-1. As expected, the results showed that SAS did increase the MKP-1 expression (Figure 4E). Considering the fact that cAMP or its analogue can up-regulate MKP-1 expression in various cells [47,48,49,50,51], we investigated the effect of SAS on the intracellular cAMP level. Indeed, SAS prevented the TNF-α-caused decrease of the cAMP level in 3T3-L1 pre-adipocytes (Figure 5A,B). Moreover, RP-cAMPS, a cAMP antagonist, could block the effect of SAS on MCP-1 (Figure 5C). As a principal effector of cAMP [52,53], PKA could also be activated by SAS in TNF-α-treated adipocytes (Figure 5E). Consistently, the PKA antagonist H89 could block the effect of SAS (Figure 5D). These findings demonstrate that SAS decreases MCP-1 production in TNF-α-activated adipocytes, primarily via the cAMP/PKA pathway.

PDE3B, an essential regulator of the cAMP level in adipocytes, plays a key role in the regulation of lipolysis, energy homeostasis and insulin secretion. Altered levels of PDE3B may result in a number of changes in the regulation of glucose and lipid metabolism and in the overall energy homeostasis [54]. In PDE3B knockout mice, the adipocyte size and the amounts of adipose tissue were decreased, and the lean mass was increased compared with the wild-type control mice [13,55]. Our result showed that two PDE3 inhibitors (milrinone and cilostamide) could suppress TNF-α-induced MCP-1 production in adipocytes (Appendix A). Thus, we further checked whether SAS targeted PDE3B in order to exert its effects. The data showed the that SAS directly bound to the PDE3B protein (Figure 6A) and inhibited its activity (Figure 6B), while the silence of PDE3B in the adipocytes could significantly decrease the TNF-α-induced MCP-1 production, and the effect of SAS on MCP-1 production was almost abolished (Figure 6C,D and Appendix A). It was consistent with previous studies that PDE3 inhibition could attenuate the recruitment of monocytes to the intestinal mucosa [56], and the ablation of PDE3B reduced the MCP-1 expression and hampered macrophage infiltration into the adipose tissue [13]. In addition to PDE3B, SAS also inhibited both the PDE3A and PDE4B activity (Figure 7), showing that it is a non-specific PDE inhibitor.

Our findings demonstrate, for the first time, that SAS is a non-selective PDE inhibitor. In TNF-α-stimulated adipocytes, it can directly inactivate PDE3B to elevate the intracellular cAMP level, which leads to the activation of PKA and subsequently increases the MKP-1 expression. The latter can dephosphorylate the activated ERK and p38, thus resulting in the decrease of MCP-1 production (Figure 8). These effects of SAS are independent of its previously identified targets, IKKβ and AMPK. This study provides the evidence for a hitherto unknown anti-inflammatory pathway of SAS in obesity-related inflammation and proposes a rationale for the application of its prodrugs in metabolic diseases.

## 4. Materials and Methods

### 4.1. Reagents

The SAS (purity > 99.5%; Figure 1) was purchased from Sinopharm Chemical Reagent Beijing Co., Ltd. (Shanghai, China). The recombinant human PDE3B was from SignalChem Lifesciences Corporation (Richmond, British Columbia, Canada). The recombinant mouse TNF-α, human PDE3A and PDE4B were obtained from Sino Biological Inc. (Beijing, China). The PDE-Glo PDE assay kit and PepTag assay kit, used for the non-radioactive detection of protein kinase A (PKA) activity, were from Promega Co. (Madison, WI, USA). The ELISA kits for the cAMP and mouse MCP-1 were from Cayman Chemical (Ann Arbor, MI, USA) and Biolegend Co. (San Diego, CA, USA), respectively. The M-MuLV First Strand cDNA Synthesis Kit was obtained from Sangon Biotech Co. (Shanghai, China). Mammalian Protein Extraction Kit was from Cwbiotech Co. (Beijing, China). ECL luminescence reagent was from Absin Bioscience Inc. (Shanghai, China). The RP-cAMPS and the monoclonal antibody for MKP-1 (E6) were from Santa Cruz Biotechnology Inc. (Dallas, TX, USA). LY2409881, H-89, Compound C, milrinone and cilostamide were purchased from MedChemExpress (Monmouth Junction, NJ, USA). The PCH biosensor for the Pioneer system was from FortéBio, LLC. (Fremont, CA, USA). The mouse IKKβ and PDE3B siRNA, and their negative control (scrambled siRNA) were designed and synthesized by GenePharma Co. (Suzhou, Jiangsu, China). The trizol reagent and the transfection reagent Entranster^TM^-R4000 were from Engreen Biosystem (Beijing, China). PVDF membrane, N-hydroxysuccinimide (NHS, CAS^#^6066-82-6), 1-ethyl-3-(3-dimethylperpyl)-carbodiimide (EDC, CAS^#^25952-53-8), insulin and IBMX were from Sigma-Aldrich (St Louis, MO, USA). The antibodies for the mouse PDE3B, GAPDH, ERK and horseradish peroxidase (HRP)-conjugated goat anti-rabbit IgG, color visualization protein ladder were from ABclonal Co. (Wuhan, Hubei, China). The monoclonal antibodies against JNK, p-JNK, p-ERK, p-p38, p38, IKKβ (D30C6) and biotinylated protein ladder were from Cell Signaling Technology Co. (Danvers, MA, USA). Rolipram, dexamethasone, rosiglitazone and sodium pyruvate were from Tokyo Chemical Industry Development Co., Ltd. (Tokyo, Japan). Collagenase I was from Yuanye Bio-Technology Co., Ltd. (Shanghai, China). The non-fat milk powder and 2,2,2-tribromoethanol were from Applygen Technologies Inc. (Beijing, China) and Aladdin Co. (Shanghai, China), respectively. All other reagents were of analytical grade.

### 4.2. Cells and Animals

The murine fibroblasts of the immortalized cell line 3T3-L1 (pre-adipocyte; passage no.15) were obtained from Cell Center of Institute of Basic Medical Sciences, Chinese Academy of Medical Sciences (Beijing, China). They were cultured in DMEM containing 10% heat-inactivated fetal calf serum in a humidified incubator with 5.0% CO_2_ at 37 °C.

The male C57BL/6N mice (18–20 g) were from Vital River Experimental Animal Services (Beijing, China) and housed in an SPF laboratory under standard temperature (22–24 °C) and humidity (45–65%) conditions with a 12 h light/dark cycle and water ad libitum. The animal study was reviewed and approved by the Institutional Care and Use Committee of the Institute of Medicinal Plant Development (IMPLAD) of Chinese Academy of Medical Sciences and carried out according to the Guidelines for the Care and Use of Laboratory Animals (NIH Publications No. 8023, revised 1978). Anesthetic drugs and all other necessary measures were used for alleviating animal suffering during the experimental procedures.

### 4.3. Isolation and Culture of Mouse Primary Mature Adipocytes

The mouse primary adipocytes were isolated from normal lean male C57BL/6N mice, as described previously, with modification [57,58]. In brief, the mice were anesthetized by an intraperitoneal injection of 2,2,2-tribromoethanol (200 mg/kg) and sacrificed via a dislocated neck. The epididymal fat pads were removed, minced and digested by collagenase I (2 mg/mL in DMEM supplemented with 20 mg/mL BSA) at 37 °C for 1 h. The digested tissue was filtered, and the filtrate was centrifuged at 200× *g* for 5 min. Remove and discard the underlying pellet (containing pre-adipocytes, fibroblasts, and erythrocytes) and media. Re-suspend the remaining fat layer in 10 mL DMEM and centrifuge at 200× *g* for 5 min. Repeat this step two additional times. The floating top layer containing primary differentiated mature adipocytes was collected and cultured in DMEM supplemented with 10% fetal bovine serum (FBS).

### 4.4. Differentiation of 3T3-L1 Adipocytes

The 3T3-L1 pre-adipocytes were cultured and differentiated into mature adipocytes, as previously described, with slight modification [59]. The cells were maintained in M1 medium (DMEM supplemented with 10% FBS, 1 mM sodium pyruvate, 4 mM glutamine and 1% penicillin-streptomycin solution) to reach confluence. Approximately 2 days later (termed day 0), the medium was replaced with M2 medium (M1 medium supplemented with 1.5 μg/mL insulin, 1 μM dexamethasone, 0.5 mM IBMX and 2 μM rosiglitazone). Two days later, the medium was then changed to M3 medium (M1 medium supplemented with 1.5 μg/mL insulin). Two days later, the medium was replaced with M2 for another 2 days. From day 6, the cells were maintained in M3 medium with a medium change every day. About 8–10 days later, a microscope could observe a lot of lipid droplets and the mature adipocytes were obtained.

### 4.5. Determination of MCP-1 Level in Supernatant or Subnatant

The adipocytes were pretreated with SAS at different concentrations (0–5 mM) for 1 h and then stimulated with mouse TNF-α (40 ng/mL) at 37 °C for 24 h. The MCP-1 level in supernatant (for 3T3-L1 adipocytes) or subnatant (for primary adipocytes) was determined by ELISA according to the manufacturer’s instruction. The concentration of MCP-1 was calculated from the standard curve.

### 4.6. RNA Extraction and Quantitative Real-Time PCR (RT-qPCR)

The adipocytes were pretreated with SAS at different concentrations (0–5 mM) for 1 h and then stimulated with mouse TNF-α (40 ng/mL) at 37 °C for 4 h. The total mRNA was isolated from 3T3-L1 pre-adipocytes by Trizol reagent according to the manufacturer’s instruction. Reverse transcription reactions were conducted by using M-MuLV first strand cDNA synthesis kit. The RT-qPCR analyses were performed on a BIOER Fluorescent Quantitative Detection System (Bioer Technology, Hangzhou, China). The procedure condition was as follows: 95 °C for 30 s, 95 °C for 5 s with 40 cycles, and 62 °C for 30 s. The comparative Ct method (2^−ΔΔCt^) was used to analyze the relative intensities. The primer sequences for MCP-1 were as follows: 5′-GCCCCACTCACCTGCTGCTACT-3′ (forward) and 5′-CCTGCTGCTGGTGATCCTCTTGT-3′ (reverse). The primer sequences for β-actin were as follows: 5′-TGTTACCAACTGGGACGACA-3′ (forward) and 5′-AAGGAAGGCTGGAAAAGAGC-3′ (reverse).

The mRNA stability assay was performed as described previously with modification [60]. The 3T3-L1 pre-adipocytes were stimulated with TNF-α (40 ng/mL) for 2 h, and then treated with actinomycin D (5 ug/mL). SAS (5 mM) was added at different time points (0, 0.5, 2, 4 h). The total RNA were harvested, and the MCP-1 mRNA levels were measured by RT-qPCR analysis.

### 4.7. Western Blot Assay

The 3T3-L1 pre-adipocytes (6 × 10^6^ cells per well in 6-well plates) were incubated with SAS for 1 h and then stimulated with mouse TNF-α (40 ng/mL) at 37 °C for indicated time. The total protein was extracted with the mammalian protein extraction kit. For western blot assay, equal amounts of protein samples (30 µg/lane) were separated by 10% sodium dodecyl sulphate-polyacrylamide gel electrophoresis (SDS-PAGE) and then transferred to PVDF membranes. Subsequently, the membranes were blocked with 5% non-fat milk powder in TBST (0.9 mM Tris, 9 mM Tris-HCl, 150 mM NaCl, 0.05% Tween-20) for 2 h at room temperature, and then incubated with various primary antibodies overnight at 4 °C. After being washed by TBST, the membranes were incubated with the corresponding secondary antibodies for 1 h at room temperature. After successive washes, the protein bands were detected using ECL luminescence reagent and photographed by the ChemiDoc XRS+ imaging system (Bio-Rad Technology, Hercules, California, America) or ChemiScope 6000 Exp (Clinx Science Instruments Co., Shanghai, China).

### 4.8. Determination of Intracellular cAMP Level

To determine the time-course change of the TNF-α-induced intracellular cAMP, the 3T3-L1 pre-adipocytes were pretreated with or without SAS (5 mM) for 1 h and then exposed to TNF-α (40 ng/mL) for different time. The cells were lysed, and the changes of intracellular cAMP within 4 h were measured by using a commercial cAMP ELISA kit. To determine the concentration-effect of SAS on intracellular cAMP, 3T3-L1 pre-adipocytes were pretreated with SAS for 1 h at the indicated concentrations and then exposed to TNF-α (40 ng/mL) for further 1 h. The cells were lysed, and the intracellular cAMP level was measured by using a commercial cAMP ELISA kit. The protein concentration of each sample was determined using BCA assay, and the cAMP level of each sample was calculated according to the below formula:cAMP level (pmol/mg protein) =
cAMP concentration (pmol/mL)/protein concentration (mg/mL)

### 4.9. PKA Activity Assay

The 3T3-L1 pre-adipocytes were pretreated with SAS for 1 h and then stimulated with TNF-α (40 ng/mL) for further 1 h. The cells were lysed by cold PKA extraction buffer [25 mM Tris-HCl (pH 7.4), 0.5 mM EDTA, 0.5 mM EGTA, 10 mM β-mercaptoethanol, 1 μg/mL leupetin, 1 μg/mL aprotinin and 0.5 mM PMSF] and centrifuged (5000× *g*) at 4 °C for 10 min. The supernatant was used immediately for PKA activity determination by using a PepTag assay for non-radioactive detection of PKA.

### 4.10. Surface Plasmon Resonance (SPR) Assay

A SPR assay was widely used for assaying intermolecular interactions in real time, including the interaction of small molecules with proteins [61,62]. In the present study, this assay was performed using a Pioneer System (Pall FortéBio, LLC.) in a running buffer PBST (137 mM NaCl, 2.68 mM KCl, 1.47 mM KH_2_PO_4_, 9.75 mM Na_2_HPO_4_, pH 7.4, 0.05% Tween 20 and 1% DMSO). The recombinant human PDE3B was immobilized onto a PCH biosensor chip using NHS (100 mM) and EDC (400 mM) coupling reagents in 10 mM sodium acetate (pH 4.0). To determine the binding affinity of SAS and PDE3B, SAS was injected into the PDE3B-immobilized channel at various concentrations. Both association value (Ka) and dissociation value (Kd) were determined.

### 4.11. Gene Silencing

The PDE3B siRNA was a pool of three PDE3B-specific siRNAs, and the IKKβ siRNA was a pool of two IKKβ-specific siRNAs. The sequences of these siRNAs were listed in Table 1. The 3T3-L1 pre-adipocytes were cultured in 6-well plates (3 × 10^6^ cells per well). Twenty-four hours later, the cells were transfected with IKKβ or PDE3B siRNA (3.33 μg/well), or the scrambled siRNA using transfection reagent Entranster^TM^-R4000 (5 μL/well) according to the manufacturer’s protocol. Six hours later, the culture medium was changed to fresh medium, and the cells were cultured for further 48 h or 72 h. The obtained target gene-deficient cells were confirmed by RT-qPCR or western blot.

### 4.12. Statistical Analysis

All of the statistical analyses were performed with the GraphPad Prism (version 7.0). Comparisons between two groups were performed using an unpaired Student’s *t*-test. Comparisons between multiple treatment groups were performed using a one-way ANOVA with the Tukey’s *post hoc* analysis. All of the data were reported as mean ± SD of at least three independent experiments. *p* < 0.05 was considered statistically significant.

## Figures and Tables

**Figure 1 ijms-24-00320-f001:**
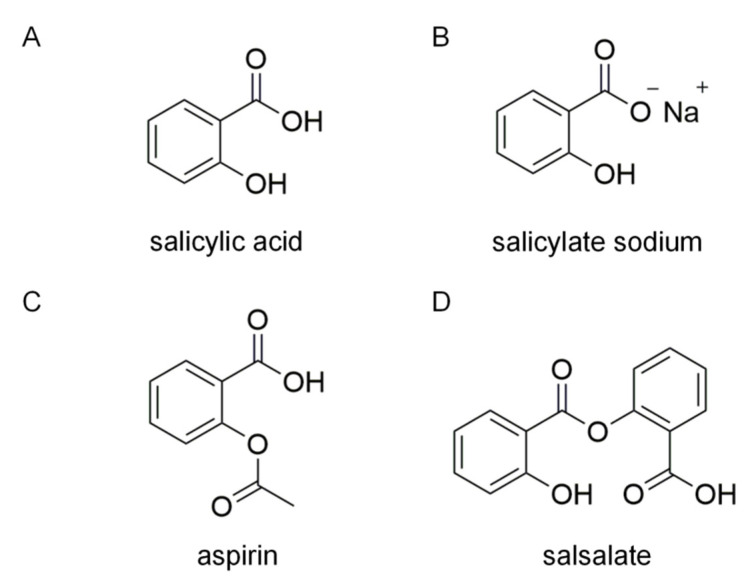
Chemical structures of salicylic acid (**A**), salicylate sodium (**B**), aspirin (**C**) and salsalate (**D**).

**Figure 2 ijms-24-00320-f002:**
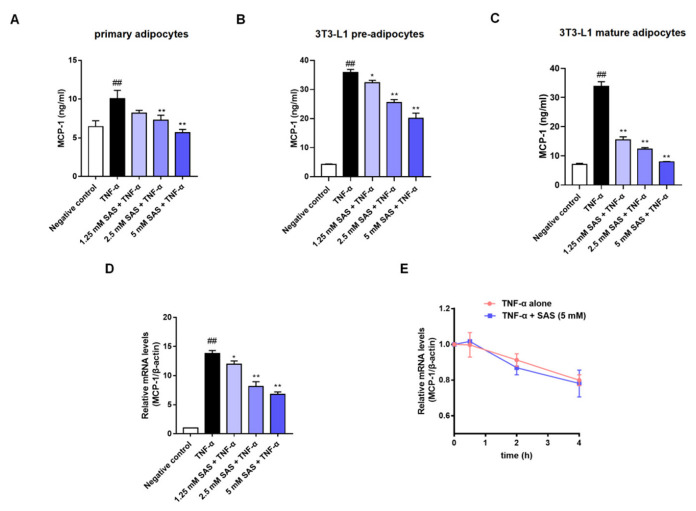
SAS decreases MCP-1 production in adipocytes. (**A**–**C**) Effects of SAS on MCP-1 production in the subnatant or supernatant of TNF-α-stimulated mouse primary adipocytes (**A**), 3T3-L1 pre-adipocytes (**B**) and 3T3-L1 mature adipocytes (**C**). (**D**) Effect of SAS on MCP-1 mRNA in TNF-α-stimulated 3T3-L1 pre-adipocytes. (**E**) Effect of SAS on the stability of MCP-1 mRNA. Data were shown as the mean ± SD (n = 3). ^##^
*p* < 0.01 vs. negative control (vehicle); * *p* < 0.05 and ** *p* < 0.01 vs. TNF-α alone.

**Figure 3 ijms-24-00320-f003:**
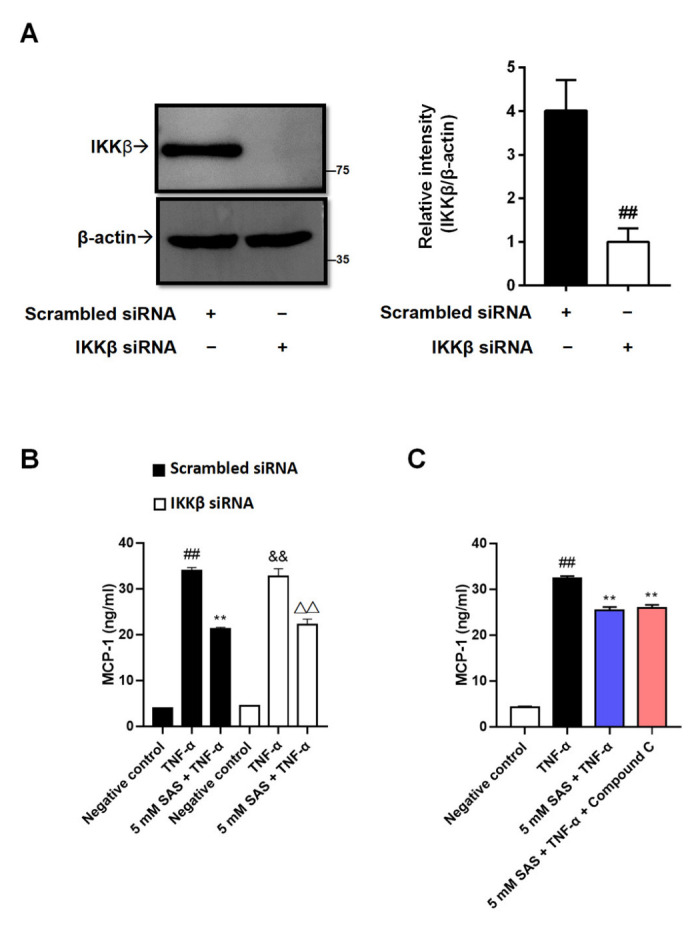
SAS decreases MCP-1 production independent of IKKβ inhibition and AMPK activation in 3T3-L1 pre-adipocytes. (**A**) IKKβ expressions in 3T3-L1 pre-adipocytes transfected with IKKβ siRNA or the scrambled siRNA. Data were shown as the mean ± SD (n = 3). ^##^
*p* < 0.01 vs. scrambled siRNA group. (**B**) SAS decreases TNF-α-induced MCP-1 production in IKKβ-deficient 3T3-L1 pre-adipocytes. Data were shown as the mean ± SD (n = 3). For scrambled siRNA group, ^##^
*p* < 0.01 vs. negative control (vehicle); ** *p* < 0.01 vs. TNF-α alone. For IKKβ siRNA group, ^&&^
*p* < 0.01 vs. negative control (vehicle); ^ΔΔ^
*p* < 0.01 vs. TNF-α alone. (**C**) SAS decreases supernatant MCP-1 in TNF-α-stimulated 3T3-L1 pre-adipocytes in the presence of AMPK inhibitor (Compound C). Data were shown as the mean ± SD (n = 3). ^##^
*p* < 0.01 vs. negative control (vehicle); ** *p* < 0.01 vs. TNF-α alone.

**Figure 4 ijms-24-00320-f004:**
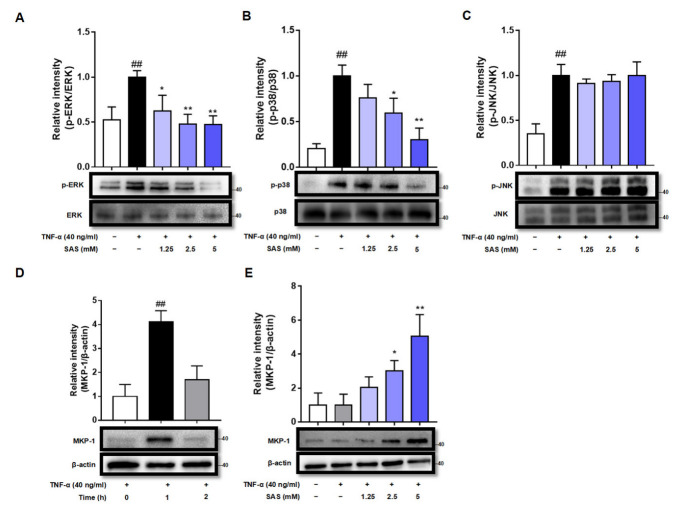
SAS reduces the levels of p-p38 and p-ERK and increases MKP-1 expression in TNF-α-stimulated 3T3-L1 pre-adipocytes. (**A**–**C**) SAS reduces the levels of p-ERK (**A**) and p-p38 (**B**), but not p-JNK (**C**) in TNF-α-stimulated 3T3-L1 pre-adipocytes. ^##^
*p* < 0.01 vs. negative control (vehicle); * *p* < 0.05 and ** *p* < 0.01 vs. TNF-α alone. (**D**) Time-course of MKP-1 expression in TNF-α-stimulated 3T3-L1 pre-adipocytes. ^##^
*p* < 0.01 vs. 0 h. (**E**) SAS concentration-dependently increases MKP-1 expression in TNF-α-stimulated 3T3-L1 pre-adipocytes. Data were shown as the mean ± SD (n = 3). * *p* < 0.05 and ** *p* < 0.01 vs. TNF-α alone.

**Figure 5 ijms-24-00320-f005:**
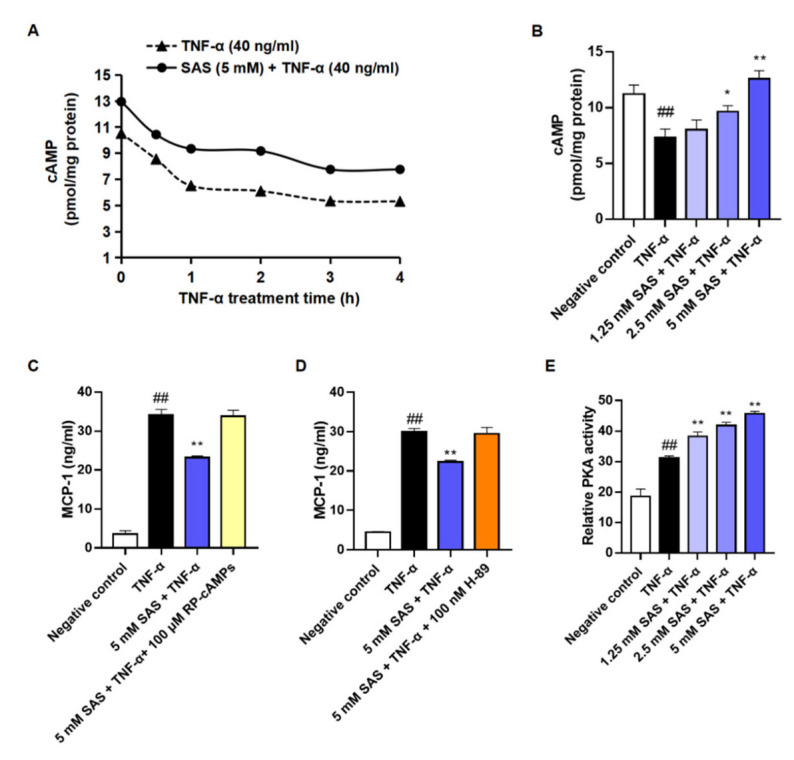
SAS decreases MCP-1 production of adipocytes depending on cAMP/PKA pathway. (**A**) Time-course change of intracellular cAMP in TNF-α-stimulated 3T3-L1 pre-adipocytes. (**B**) Effect of SAS on intracellular cAMP in TNF-α-stimulated 3T3-L1 pre-adipocytes. (**C**) RP-cAMPS (a cAMP antagonist, 100 μM) blocks the effect of SAS on supernatant MCP-1 in TNF-α-stimulated 3T3-L1 pre-adipocytes. (**D**) H-89 (a PKA antagonist, 100 nM) blocks the effect of SAS on supernatant MCP-1 in TNF-α-stimulated 3T3-L1 pre-adipocytes. (**E**) SAS increases intracellular PKA activity in the TNF-α-stimulated 3T3-L1 pre-adipocytes. Data were shown as the mean ± SD (n = 3). ^##^
*p* < 0.01 vs. negative control (vehicle); * *p* < 0.05 and ** *p* < 0.01 vs. TNF-α alone.

**Figure 6 ijms-24-00320-f006:**
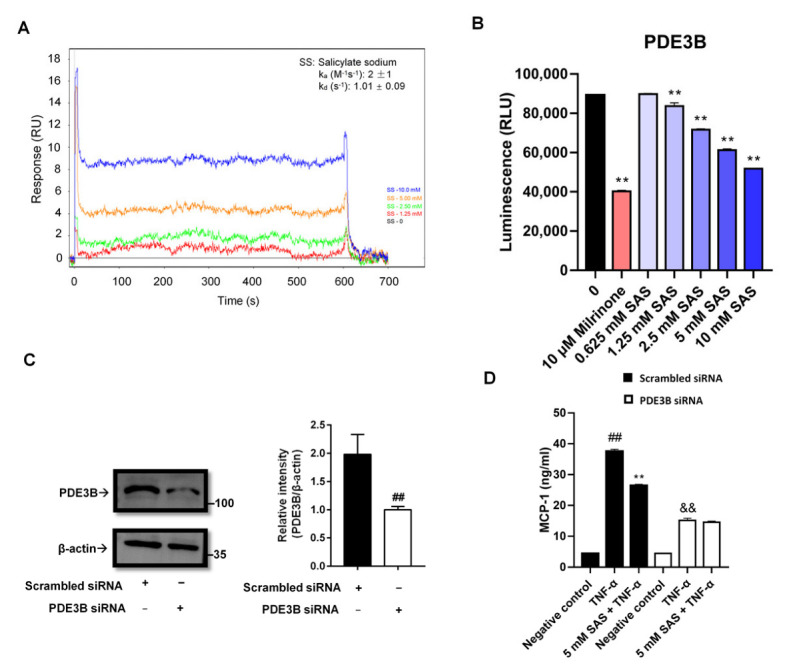
SAS directly inhibits PDE3B activity to decrease MCP-1 production in adipocytes. (**A**) SAS directly binds to PDE3B. In the SPR assay, PDE3B-immobilized biosensor chip was exposed to various concentrations of SAS ranged from 1.25 to 10 mM. The changes in response units were shown. (**B**) SAS directly inhibits PDE3B activity. Milrinone was used as a positive control. ^**^
*p* < 0.01 vs. vehicle (0 group). (**C**) PDE3B expression in 3T3-L1 pre-adipocytes transfected with PDE3B siRNA or the scrambled siRNA. (**D**) SAS fails to suppress MCP-1 production in PDE3B-deficient 3T3-L1 pre-adipocytes. Data were shown as the mean ± SD (n = 3). For scrambled siRNA group, ^##^
*p* < 0.01 vs. negative control (vehicle); ** *p* < 0.01 vs. TNF-α alone. For PDE3B siRNA group, ^&&^
*p* < 0.01 vs. negative control (vehicle).

**Figure 7 ijms-24-00320-f007:**
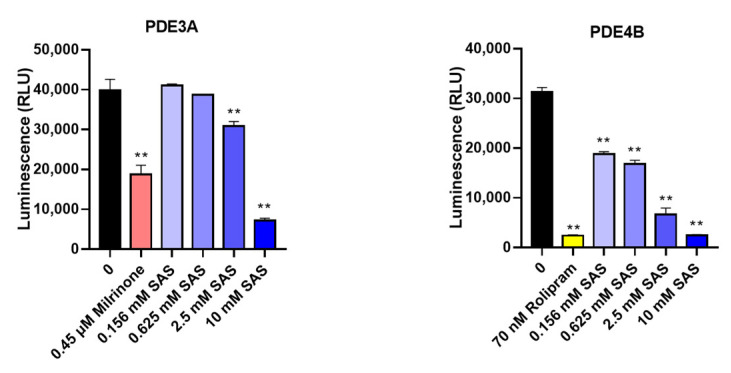
SAS directly inhibits PDE3A and PDE4B activity. Milrinone and rolipram were used as the positive controls. ** *p* < 0.01 vs. vehicle (0 group).

**Figure 8 ijms-24-00320-f008:**
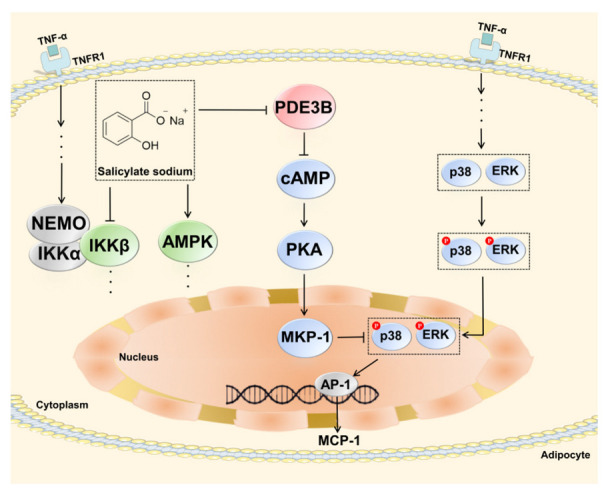
Schematic diagram depicting how SAS decreases MCP-1 production in TNF-α-stimulated adipocytes.

**Table 1 ijms-24-00320-t001:** Sequences of mouse IKKβ, PDE3B siRNA and their negative control (scrambled siRNA).

siRNA Name	Strand	Sequence (5′ to 3′)
PDE3B-1	sense	CCU CAU CGU UGC UGA CUA ATT
	antisense	UUA GUC AGC AAC GAU GAG GTT
PDE3B-2	sense	CCA UUG AAC AUU GUG GAA ATT
	antisense	UUU CCA CAA UGU UCA AUG GTT
PDE3B-3	sense	CCA UGA AAC GGA AAC CAA ATT
	antisense	UUU GGU UUC CGU UUC AUG GTT
IKKβ-1	sense	GUG AAC AGA UCG CCA UCA ATT
	antisense	UUG AUG GCG AUC UGU UCA CTT
IKKβ-2	sense	GGA CAU CGU UGU UAG UGA ATT
	antisense	UUC ACU AAC AAC GAU GUC CTT
Scrambled siRNA	sense	UUC UCC GAA CGU GUC ACG UTT
	antisense	ACG UGA CAC GUU CGG AGA ATT

## Data Availability

The datasets analyzed during the current study are available in the [Mendeley] repository, [https://data.mendeley.com/drafts/6vwz4sk4rn].

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
