# Peer review of "Salicylate Sodium Suppresses Monocyte Chemoattractant Protein-1 Production by Directly Inhibiting Phosphodiesterase 3B in TNF-α-Stimulated Adipocytes"

_ijms, 2022, doi:10.3390/ijms24010320_

Round 1
Reviewer 1 Report
The manuscript entitled “ Salicylate suppresses monocyte chemoattractant protein-1 production by targeting phosphodiesterase 3B inhibition in TNF-α-stimulated adipocytes” is of interest to scientists working in this field. The article is presented well and can be published in the present form.
I would recommend the authors to give the structures of salicylic acid derivatives that are mentioned in the text: salicylic acid, aspirin, salicylate anion (or sodium salicylate), and also salsalate - for a better understanding the relationship between these structures.
In addition, I would like to note that salicylate is a salt (or ester) of salicylic acid, in which, along with the salicylate anion, there is a cation that can also affect biological aspects. Therefore it should be indicated that sodium salicylate was used in the work at least in the title and in the keywords
Author Response
Response to Reviewer 1 Comments
Dear reviewer,
Thank you for the kind assistance and the critical review to improve our manuscript entitled "Salicylate suppresses monocyte chemoattractant protein-1 production by targeting phosphodiesterase 3B inhibition in TNF-α-stimulated adipocytes". Thank you for the recognition of our work. We have carefully studied all comments and made substantial revisions. An editable version of the article has been uploaded. In this revised version, all changes were marked up using the “Track Changes” function.
Response to the comments point to point
Point 1: I would recommend the authors to give the structures of salicylic acid derivatives that are mentioned in the text: salicylic acid, aspirin, salicylate anion (or sodium salicylate), and also salsalate - for a better understanding the relationship between these structures.
Response: Thanks to the valuable suggestion. We have supplemented the structures of these four compounds in the revised manuscript (Figure 1).
Point 2: In addition, I would like to note that salicylate is a salt (or ester) of salicylic acid, in which, along with the salicylate anion, there is a cation that can also affect biological aspects. Therefore, it should be indicated that sodium salicylate was used in the work at least in the title and in the keywords.
Response: Thanks to the valuable suggestion. We have used “salicylate sodium (SAS)” instead of “salicylate” in the right places of the manuscript, including all titles. Of note, because the title words should not be repeated in Keywords, the original Keyword “salicylate” has been deleted.
Reviewer 2 Report
Dear Author,
The topic is highly relevant. It has been suggested that salicylate can directly bind phosphodiesterase 3B (PDE3B) and decrease the level of monocytes chemoattractant protein-1 (MCP-1) in TNF-alfa stimulated adipocytes. Moreover, PDEE3A and PDE4B activity was also suppressed by salicylate.
Below, please find minor suggestions:
KEYWORDS: Salicylate; Adipocyte; MCP-1; PDE3B; cAMP (The title words should not be repeated in Keywords).
INTRODUCTION:
Line 50-51: In turn the released TNF-α in turn stimulates adipocytes to produce more free fatty acid and MCP-1.
Line 55-56: As Like the cAMP- and cGMP-specific PDE, PDE3 family consists of two members (PDE3A and PDE3B) which are generated from two similarly organized genes.
Line 57-58: In contrast to PDE3A, PDE3B is relatively more highly expressed in insulin-sensitive cells, such as adipocytes, hepatocytes, and pancreatic β cells [9, 10].
Line 60-62: Emerging evidences showed that PDE3B is also involved in inflammation-related events, including regulating NLRP3 inflammasome of adipose tissue [13], mediating rheumatoid arthritis [14] and allergic airway inflammation [15]. Line 64-65: Salicylic acid, a main metabolite of aspirin, is a plant product whose anti-inflammatory activity was firstly described by the famous Greek famous physician Hippocrates.
Line 65-66: Unlike aspirin, salicylate does not possess the acetylating activity to inactivate cyclooxygenase in vitro [16], but both salicylate and aspirin inhibit IKKβ activation [17, 18] and activate AMP-activated protein kinase (AMPK) [19].
RESULTS Line 90-91: Thus, 3T3-L1 pre-adipocytes were used for the most of the subsequent experiments. By using RT-qPCR assay, MCP-1 mRNA level was measured. Line Line 103-105: Previous As previous study reported that TNF-α-induced MCP-1 could be mediated by NF-κB [27], together with the fact that salicylate is an IKKβ inhibitor [17], we next evaluated whether salicylate reduced MCP-1 via IKKβ inhibition. Line 112-114: Different from previous report In contrast to previous reports [30], our results indicated that AICAR, an AMPK activator, did not decrease MCP-1 production in TNF-α- stimulated 3T3-L1 pre-adipocytes (Figure S4)
Author Response
Response to Reviewer 2 Comments
Dear reviewer,
Thank you for the kind assistance and patient review to improve our manuscript entitled "Salicylate suppresses monocyte chemoattractant protein-1 production by targeting phosphodiesterase 3B inhibition in TNF-α-stimulated adipocytes". Thank you for the recognition of our work. We have carefully studied all comments and made substantial revisions. An editable version of the article has been uploaded. In this revised version, all changes were marked up using the “Track Changes” function.
Response to the comments point to point
Point 1: KEYWORDS: Salicylate; Adipocyte; MCP-1; PDE3B; cAMP (The title words should not be repeated in Keywords).
Response 1:Thanks to the valuable suggestion. We have re-selected the keywords in the revised manuscript (line 31).
Point 2:
INTRODUCTION:
Line 50-51: In turn the released TNF-α in turn stimulates adipocytes to produce more free fatty acid and MCP-1.
Line 55-56: As Like the cAMP- and cGMP-specific PDE, PDE3 family consists of two members (PDE3A and PDE3B) which are generated from two similarly organized genes.
Line 57-58: In contrast to PDE3A, PDE3B is relatively more highly expressed in insulin-sensitive cells, such as adipocytes, hepatocytes, and pancreatic β cells [9, 10].
Line 60-62: Emerging evidences showed that PDE3B is also involved in inflammation-related events, including regulating NLRP3 inflammasome of adipose tissue [13], mediating rheumatoid arthritis [14] and allergic airway inflammation [15].
Line 64-65: Salicylic acid, a main metabolite of aspirin, is a plant product whose anti-inflammatory activity was firstly described by the famous Greek famous physician Hippocrates.
Line 65-66: Unlike aspirin, salicylate does not possess the acetylating activity to inactivate cyclooxygenase in vitro [16], but both salicylate and aspirin inhibit IKKβ activation [17, 18] and activate AMP-activated protein kinase (AMPK) [19].
RESULTS Line 90-91: Thus, 3T3-L1 pre-adipocytes were used for the most of the subsequent experiments. By using RT-qPCR assay, MCP-1 mRNA level was measured.
Line 103-105: Previous As previous study reported that TNF-α-induced MCP-1 could be mediated by NF-κB [27], together with the fact that salicylate is an IKKβ inhibitor [17], we next evaluated whether salicylate reduced MCP-1 via IKKβ inhibition.
Line 112-114: Different from previous report In contrast to previous reports [30], our results indicated that AICAR, an AMPK activator, did not decrease MCP-1 production in TNF-α- stimulated 3T3-L1 pre-adipocytes (Figure S4)
Response 2: We appreciate your patient check for our manuscript. Besides the sentences which had been pointed out above, we also have carefully reviewed the full text again and improved the language.
Reviewer 3 Report
Zhang et al. investigated the effect of salicylate on the TNF alpha-induced MCP-1 expression in adipocytes. They found a dose-dependent inhibition by salicylate at both protein and mRNA levels. Though salicylate was previously shown to inhibit IKK beta and AMPK, the authors elegantly demonstrate that the target of salicylate is PDB3 in adipocytes leading to enhanced MKP-1 expression and consequently to a decreased p38 and ERK phosphorylation.
Major point:
The authors state by detecting the decreased levels of the TNFa-induced m RNA expression in the presence of salicylate that this implies transcriptional regulation. However, the mRNA levels of many cytokines including MCP-1 is also regulated post-transcriptionally.
https://www.ncbi.nlm.nih.gov/pmc/articles/PMC3976576/
Thus, the stability of mRNA of MCP-1 mRNA should be also measured following TNF alpha alone and with salicylate treatments, especially that the p38 MAPK pathway was shown to regulate the stability of MCP-1 mRNA.
Minor points:
Several sentences in the text are superficial or grammatically incorrect. The language of the paper should be improved.
Examples:
These clinical reports suggest that salicylate plays an important role in IR-related adipose inflammation. Should be adipose tissue. Salicylate has no important role in it, maximum it can interfere with it or affect it when it is administered.
These findings indicate that salicylate suppresses MCP-1 production through targeting PDE3B inhibition…
Correctly through targeting PDE3B leading to its inhibition etc.
Author Response
Response to Reviewer 3 Comments
Dear reviewer,
Thank you for the kind assistance and the critical review to improve our manuscript entitled "Salicylate suppresses monocyte chemoattractant protein-1 production by targeting phosphodiesterase 3B inhibition in TNF-α-stimulated adipocytes". Thank you for the recognition of our work. We have carefully studied all comments and made substantial revisions. An editable version of the article has been uploaded. In this revised version, all changes were marked up using the “Track Changes” function.
Response to the comments point to point
Point 1:
- Major point:
The authors state by detecting the decreased levels of the TNFa-induced mRNA expression in the presence of salicylate that this implies transcriptional regulation. However, the mRNA levels of many cytokines including MCP-1 is also regulated post-transcriptionally.
https://www.ncbi.nlm.nih.gov/pmc/articles/PMC3976576/
Thus, the stability of mRNA of MCP-1 mRNA should be also measured following TNF alpha alone and with salicylate treatments, especially that the p38 MAPK pathway was shown to regulate the stability of MCP-1 mRNA.
Response 1: We appreciate you for the valuable suggestion. We have determined the effects of salicylate sodium on the stability of mRNA of MCP-1 mRNA, and the result showed that SAS did not affect the mRNA stability of MCP-1. We have presented this data in Figure 2E (lines 100 - 101, 104) and also discussed this content (lines 257 - 261).
Point 2:
- Minor points:
Several sentences in the text are superficial or grammatically incorrect. The language of the paper should be improved.
Examples:
- These clinical reports suggest that salicylate plays an important role in IR-related adipose inflammation. Should be adipose tissue. Salicylate has no important role in it, maximum it can interfere with it or affect it when it is administered.
Response 2: Thank you for the reminding. We have corrected this sentence in the revised manuscript (lines 17, 46, 74, 77, 89).
Point 3:
(2) These findings indicate that salicylate suppresses MCP-1 production through targeting PDE3B inhibition. Correctly through targeting PDE3B leading to its inhibition etc.
Response 3: We have corrected this sentence in the manuscript (lines 3 - 4, 191, 206). Moreover, the whole manuscript has been carefully re-checked, and the language has been improved.
Round 2
Reviewer 3 Report
I accept in the present form of the manuscript, but
Do some minor text editing on the proof file!
Like lane 24 p38. one dot not two
Lane 49. In turn,
Lane 96 Since SAS did not affect the mRNA stability of MCP-1 (Figure 96
2E), our results suggest that ...